# Intraoperative cell-salvaged versus allogeneic red blood cell transfusions in high-bleeding-risk cardiovascular surgery: Protocol for a single-center, randomized, parallel-group, noninferiority trial

Akito Tsukinaga[1,2]*, Kenji Yoshitani[1,2], Soshiro Ogata[3], Takahiro Mihara[4,5], Shin Ito[6], Hiroko Kanazawa[1], Akira Shimokawa[1], Sho Masuda[1], Masahiro Morinaga[1], Yoshiaki Ito[1], Shinnosuke Miura[1], Naonori Kawamoto[7], Yosuke Inoue[8], Satsuki Fukushima[7], Hitoshi Matsuda[8], Takuma Maeda[1]

1 Department of Anesthesiology, National Cerebral and Cardiovascular Center, Suita, Osaka, Japan, 2 Department of Transfusion, National Cerebral and Cardiovascular Center, Suita, Osaka, Japan, 3 Department of Preventive Medicine and Epidemiology, National Cerebral and Cardiovascular Center, Suita, Osaka, Japan, 4 Department of Health Data Science, Graduate School of Data Science, Yokohama City University, Yokohama, Kanagawa, Japan, 5 Department of Anesthesiology, Yokohama City University School of Medicine, Yokohama, Kanagawa, Japan, 6 Department of Clinical Medicine and Development, National Cerebral and Cardiovascular Center, Suita, Osaka, Japan, 7 Department of Cardiac Surgery, National Cerebral and Cardiovascular Center, Suita, Osaka, Japan, 8 Department of Cardiovascular Surgery, National Cerebral and Cardiovascular Center, Suita, Osaka, Japan

* akito.tsukinaga30303@gmail.com

## Abstract

### Introduction

Patients who undergo cardiac surgery have a high risk of significant blood loss and anemia. While allogeneic red blood cell (RBC) transfusions are effective, they are associated with increased morbidity and mortality. Intraoperative cell salvage may reduce reliance on allogeneic transfusions. Although intraoperative RBC salvage is recommended for cardiovascular surgery under cardiopulmonary bypass, there are concerns about increased bleeding due to coagulopathy in patients with a high bleeding risk, and its clinical impact remains unclear. This study aims to evaluate whether salvaged RBC transfusion is noninferior to allogeneic transfusion in terms of postoperative blood loss in patients with a high bleeding risk.

### Methods

This single-center, randomized, two-arm, parallel group, interventional clinical trial will include 142 participants aged ≥ 40 years with a high bleeding risk who undergo elective cardiovascular surgery with cardiopulmonary bypass. Participants will be randomly assigned to receive either salvaged RBC or allogeneic RBC transfusions intraoperatively. The primary outcome is postoperative chest tube blood loss within 12 hours from the end

**Data availability statement:** Deidentified participant-level data underlying the results reported in this article will be deposited in the Dryad digital repository (https://datadryad.org) upon completion of the trial and publication of the primary results. Access to the dataset will be unrestricted to facilitate verification, replication, and secondary analyses by other researchers.

**Funding:** The specific funding for this study was internally sourced from the National Cerebral and Cardiovascular Center. The funders (National Cerebral and Cardiovascular Center) had no role in study design, data collection and analysis, decision to publish, or preparation of the manuscript.

**Competing interests:** The authors have declared that no competing interests exist.

of surgery. Noninferiority will be demonstrated if the upper limit of the 95% confidence interval for the mean difference in blood loss between the salvaged and allogeneic groups is <200 mL. Secondary outcomes comprise the RBC transfusion volume intraoperatively and for 12 hours from the end of surgery, prevalence of re-thoracotomy within 48 hours from the end of surgery, and prevalence of ≥ 1000 mL postoperative chest tube blood loss within 12 hours from the end of surgery. These outcomes will be analyzed using the modified intention-to-treat set and repeated, for sensitivity reasons, for the per-protocol set.

## Discussion

Our trial aims to determine the noninferiority of intraoperative RBC salvage compared with allogeneic blood transfusions regarding postoperative blood loss and to promote its use in surgical procedures with a high bleeding risk.

## Trial registration

The trial was registered with the Japan Registry of Clinical Trials (jRCT1052240102) on July 30, 2024.

## Introduction

Patients who undergo cardiac procedures have a high risk of blood loss and severe anemia, necessitating transfusion. While allogeneic red blood cell (RBC) transfusions are effective, they are associated with increased postoperative morbidity and mortality [1–4]. Interventions, such as intraoperative cell salvage, have been proposed to minimize reliance on allogeneic transfusions. Interventions to reduce allogeneic blood exposure in patients who undergo planned surgery may help conserve blood stock, reduce costs, and mitigate risk to patients.

One means of reducing allogeneic RBC transfusions is cell salvage. Intraoperative cell salvage is the method of harvesting RBCs shed during surgery, and enriching and washing them for safe return to the patient's circulation as a salvaged RBC transfusion during or immediately after surgery. Intraoperative cell salvage during cardiovascular surgery under cardiopulmonary bypass (CPB) is strongly recommended to conserve allogeneic RBC transfusion stocks [5,6]. However, this recommendation is limited to patients without a high risk of bleeding and requiring only a relatively small amount of washed salvaged RBC transfusion. High-volume salvaged RBC transfusions may lead to coagulopathy caused by hemodilution or residual heparin that could not be completely removed by the washing process [7]. The hematocrit of salvaged RBC solutions [8] is slightly lower than or comparable to that of packed allogeneic RBC solutions [9,10], and it remains unclear whether they cause dilutional coagulopathy. Even if heparin residue is present, its clinical effects can potentially be neutralized by monitoring heparin concentration and counteracting its effects with protamine. Moreover, no high-quality studies have compared clinical outcomes after salvaged RBC versus

allogeneic RBC transfusions in patient populations with a high risk of bleeding. The purpose of this study is to investigate whether salvaged RBC transfusions can be used safely compared with allogeneic RBC transfusions in patients with a high risk of bleeding.

## Materials and methods

### Participants, interventions, and outcomes

**Trial design.** This is a single-center, randomized, two-arm, parallel group, controlled, outcome observer-, surgeon-, postoperative caregiver-, and participant-blinded noninferiority interventional clinical trial. Block randomization of the participants, stratified by the presence or absence of aortic arch replacement with selective cerebral perfusion, and age 40–64 or ≥ 65 years, with a 1:1 allocation will be performed.

Our protocol follows the Standard Protocol Items: Recommendations for Interventional Trials (SPIRIT) guidelines (S1 File) [11].

**Study setting.** This study will be performed at the National Cerebral and Cardiovascular Center in Japan. This hospital is the leading facility in the country, performing 1363 adult cardiovascular surgeries in 2023 (https://www.ncvc.go.jp/hospital/about/results/#shinzo_k).

**Eligibility criteria for the participants.** In this study, eligibility will be determined in two stages: provisional registration and final registration. The inclusion criteria for provisional registration are as follows:

1) Age ≥ 40 years

2) Elective cardiovascular surgery via median sternotomy with CPB with a high risk of bleeding defined by any of the following:

   a) History of previous cardiac surgery with median sternotomy ("re-do")

   b) Aortic root surgery (Ross procedure, aortic valve replacement with annular enlargement, aortic root replacement), or ascending aorta or aortic arch replacement

   c) Valvular surgery of two or more valves, excluding tricuspid annuloplasty

   d) Coronary artery bypass surgery combined with valvular surgery, excluding tricuspid annuloplasty.

The exclusion criteria for provisional registration are as follows:

1) Replacement of the descending or thoracoabdominal aorta, heart transplantation, ventricular assist device implantation, or pulmonary valve replacement

2) Extremely high risk of bleeding, defined as patients with three or more prior cardiovascular surgeries via median sternotomy. For other cases in which the bleeding risk is considered to be extremely high, the final decision on exclusion will be made through discussion between the surgeon and the anesthesiologist.

3) Antiplatelet agent or anticoagulant use that does not comply with the following withdrawal periods:

   a) aspirin, prasugrel, ticlopidine: 7 days

   b) clopidogrel: 5 days

   c) cilostazol, ticagrelor: 3 days

   d) warfarin, direct oral anticoagulants (dabigatran etexilate, rivaroxaban, apixaban, edoxaban): 3 days

4) Active bacterial or viral infections

5) Pregnant or breastfeeding

6) Rh(D) antigen negativity

7) Irregular RBC antibody positivity

8) Patients judged inappropriate for participation by the researchers.

The final registration inclusion criterion is patients who receive allogeneic RBC products during CPB. This criterion is established to exclude surgeries that are not associated with a high risk of bleeding. In these procedures, anemia can be improved solely by transfusing a small amount of salvaged blood without allogeneic RBC transfusion.

The exclusion criteria for final registration are as follows:

1) Failure to comply with the antiplatelet or anticoagulant withdrawal period(s) between provisional registration and the date of surgery

2) Change in the surgical technique between provisional registration and the date of surgery, or during surgery, and the procedure conflicts with the inclusion criteria or meets any exclusion criteria for provisional registration.

**Interventions.** Written informed consent will be obtained, and patients will be provisionally registered by the day of surgery. Final registration and allocation will be performed until just before weaning from CPB. The study product will be transfused from CPB weaning until the participant leaves the operating room if anemia, defined as hemoglobin < 100 g/L, is confirmed by laboratory testing or anticipated. This liberal transfusion threshold was chosen to prioritize participant safety, as the study targets participants at high risk of bleeding. Because perioperative anemia can lead to adverse outcomes [12], this threshold helps avoid unintended anemia and ensures a stable clinical condition for evaluating the safety and efficacy of salvaged blood. In the intervention group, salvaged RBCs will be transfused preferentially to allogeneic RBCs. If the anemia has not been corrected after salvaged RBC transfusion, allogeneic RBCs will be transfused. If salvaged RBCs are generated during allogeneic RBC transfusion, transfusion of allogeneic RBCs will be stopped, and salvaged RBC transfusion will be initiated. When there are no available salvaged RBCs, allogeneic RBCs will be transfused for anemia. In the control group, allogeneic RBCs will be transfused exclusively to treat anemia.

In accordance with the instruction manual, we will prepare the Cell Saver Elite+ (Haemonetics, Boston, MA, USA) and install the cell salvage disposables in all cases, regardless of allocation to the intervention or control group, to perform the intraoperative cell salvage procedure. The suction level will be automatically regulated using SmartSuction mode, set between 20 mmHg and 150 mmHg, to optimize fluid removal. In cases of rapid bleeding, it will be permitted to temporarily increase the pressure to 250 mmHg in manual mode. The reservoir will be primed with heparinized saline, prepared by mixing 15,000 units of unfractionated heparin with 500 mL of normal saline. This solution will be administered at approximately 15 mL per 100 mL of collected blood entering the reservoir by adjusting the roller clamp on the anticoagulant line. Other settings will be as follows:

• Fat Reduction: Off

• Auto-Wash: On

• Partial Bowl Wash: Ask

• Pump Regulation: On

• Smart Empty: On

• Centrifuge bowl size: 225 mL

• Fill Pump Rate: 500 mL/min

- Minimum Wash Volume: 1000 mL

- Wash Pump Rate: 450 mL/min

Blood from the surgical field will be suctioned into the Cell Saver Elite+ device. During CPB—from the point where unfractionated heparin is administered and the activated clotting time (ACT) exceeds 200 seconds until protamine is administered—blood will be suctioned into the CPB circuit. After separation of the participant from CPB, the remaining blood in the CPB circuit will be promptly collected into the Cell Saver Elite+ device. In the control group, the same collection, washing, and storage in the reinfusion bag will be performed as in the intervention group, in order to preserve blinding.

**Interventions – modifications.**  The following conditions will result in the discontinuation of the experimental treatment:

1)  If pus or other material, which causes new infection due to salvaged RBC transfusion, is identified

2) In the allogeneic blood group, if the anesthesiologist determines that the participant may develop critical anemia if the salvaged RBCs are not used when the availability of allogeneic blood products is low

3) If the surgical procedure is changed to one with an extremely high risk of bleeding owing to unexpected complications during surgery

4) If circulatory mechanical support is required during surgery

5) If the principal investigator or sub-investigator determines that it is difficult to continue the study.

**Intervention – concomitant care.**  Treatment for preoperative anemia, including RBC transfusion, is not protocol-defined; such decisions will be left to the discretion of the attending physician on the basis of the clinical context. For intraoperative management, hydroxyethyl starch 130/0.4 (Voluven; Fresenius/Hospira, Germany) can be infused up to 50 mL/kg during surgery and cannot be infused after surgery and during re-thoracotomy. Tranexamic acid will be infused at anesthesia induction at a dose of 1 g and by continuous infusion at a rate of 2 mg/kg/h until the end of surgery. We will administer 300–400 U/kg of unfractionated heparin to reach an ACT of > 400 seconds and administer additional heparin as needed to maintain the target ACT. If the ACT does not reach > 400 seconds, 500–1500 U of antithrombin 3 will be infused. For cardiac surgeries, the CPB circuit will be primed with 1500 mL of crystalloid solution. In major vascular surgeries involving selective cerebral perfusion, an additional 200–300 mL of mannitol will be added to the priming solution. A 100 U/kg dose of unfractionated heparin will be administered in the CPB prime solution. During CPB, 100 U/kg of unfractionated heparin will be administered and ACT will be monitored every hour. Participants who require selective antegrade cerebral perfusion will be cooled until the nasopharyngeal temperature reaches 23–28°C. The temperature during CPB for other participants will be targeted at approximately 34°C. Blood in the surgical field will be aspirated into the CPB reservoir, but it is acceptable at the surgeon's discretion that blood will be aspirated into the Cell Saver Elite+. All blood in the Cell Saver Elite+ will be washed and returned to the participant during CPB. RBC transfusion will be performed to maintain a hemoglobin concentration of > 80 g/L and managed to reach approximately 100 g/L at the time of weaning from CPB. Fresh-frozen plasma (FFP) will not be administered in the CPB circuit.

After separation from CPB, unfractionated heparin will be reversed with protamine at 3 mg/kg. We will administer further protamine injections at 20–50 mg per dose on the basis of the difference between heparinase-treated ACT and standard ACT > 20 seconds or clot time ratio > 1.2. Cryoprecipitate will always be ordered when fibrinogen concentration is < 1 g/L as measured before CPB weaning, or 1–1.5 g/L at the physician's discretion, and not ordered when the concentration is > 1.5 g/L. If ordered, cryoprecipitate will be administered after completion of protamine reversal and achievement of surgical hemostasis. FFP will be initiated at CPB weaning and administered in two scenarios: in cases of massive bleeding

when laboratory test results are not yet available due to turnaround time, FFP will be transfused with a target ratio of approximately 1:1 to RBCs. In other cases, FFP will be administered when the fibrinogen concentration is < 2.0 g/L or when clinical bleeding persists despite appropriate heparin reversal and platelet transfusion for thrombocytopenia. Platelet transfusion will be initiated after completion of protamine reversal and achievement of surgical hemostasis. Platelet concentrates are preordered on basis of the surgical procedure and preoperative or intra-CPB platelet counts, aiming to maintain platelet counts between 50 and $100 \times 10^9$/L. If bleeding persists and platelet dysfunction is clinically suspected, platelet transfusion will be administered, in accordance with the Japanese national guidelines, with the goal of raising the platelet count to ≥ $100 \times 10^9$/L [13].

After surgery, participants will be transferred to the intensive care unit (ICU) for monitoring and mechanical ventilation; participants will be extubated when standard criteria are met. Only an allogeneic RBC product will be used for anemia after arrival at the ICU in both groups. Salvaged RBCs will never be transfused for postoperative anemia. FFP will be transfused when activated partial thromboplastin time is > 50 seconds, prothrombin time/international normalized ratio is > 1.3, or fibrinogen concentration is < 1.5 g/L, and there is a clinical bleeding tendency. In the case of massive bleeding, the ratio of FFP transfusion to allogeneic RBC transfusion in the ICU will equal approximately 1. Platelet transfusion will be administered when the platelet count is < $100 \times 10^9$/L and there is a clinical bleeding tendency. For anemia during re-thoracotomy, only allogeneic blood will be transfused; salvaged RBCs will never be transfused. The transfusion criteria for other blood products during re-thoracotomy are the same as those used during the initial surgery.

**Intervention – Laboratory and viscoelastic testing.** To guide transfusion management and assess coagulation status, we will perform standard laboratory tests, point-of-care tests such as blood gas analysis and ACT, as well as viscoelastic testing at specific perioperative time points. Laboratory tests will include complete blood count and coagulation parameters such as prothrombin time, activated partial thromboplastin time, and fibrinogen concentration. These tests will be performed during CPB, after CPB weaning (following protamine administration), and every hour thereafter. In the ICU, these tests will be routinely performed at ICU admission, at approximately 10 pm on the day of surgery, and approximately 6 am on the following day. However, the timing and frequency may be adjusted at the discretion of the attending physician.

Blood gas analysis and ACT measurements will be performed at anesthesia induction, after full systemic heparinization, at initiation and before termination of CPB, every hour during CPB, after CPB weaning (following protamine administration), and every hour thereafter. In the ICU, blood gas analysis will be performed at ICU admission and every 2 hours until 9 am on the following day, with adjustments permitted on basis of clinical judgment. ACT will be measured using the ACT plus® analyzer (Medtronic, Minneapolis, MN, USA). The High Range ACT cartridge will be used until CPB weaning, and the Heparinase Test Cartridge will be used thereafter to guide heparin reversal and detect heparin rebound.

Viscoelastic testing using the Quantra® Hemostasis Analyzer (HemoSonics, LLC, Charlottesville, VA, USA) will be performed after CPB weaning (following protamine administration), every hour thereafter, and upon ICU admission. This device is based on sonic estimation of elasticity via resonance (sonorheometry), a technology that uses ultrasound to assess changes in the shear modulus of blood—a direct physical measure of clot stiffness—during testing [14,15]. Quantra measures clot time, clot time with heparinase, clot time ratio, clot stiffness, platelet contribution to clot stiffness, and fibrinogen contribution to clot stiffness. Among these parameters, clot time ratio is significantly correlated with anti-Xa activity (Spearman coefficient 0.95; p = 0.0012) and is used intraoperatively to detect residual heparin effect and guide additional protamine administration [16]. The other parameters are included as exploratory outcomes to help elucidate potential mechanisms of bleeding, such as platelet dysfunction (platelet contribution to clot stiffness) or fibrinogen deficiency (fibrinogen contribution to clot stiffness).

**Interventions—adherence.** Protocol violations related to the administration of study blood products, as well as pre-, concomitant, or posttreatment procedures, are anticipated in approximately 5% of cases. The most likely violation is transfusion despite the absence of predefined transfusion thresholds. To promote adherence, we will provide

standardized explanations of the study protocol to all relevant medical staff prior to each surgical case, ensuring a shared understanding of transfusion thresholds, hemostatic management, and individual responsibilities. To further minimize violations, we will perform regular investigator meetings and continuously monitor study data to ensure consistent protocol implementation. Any violations will be documented, reviewed, and addressed in a timely manner. Adherence is expected to remain high; participants with major protocol violations will be excluded from the per-protocol analysis set, as outlined in the Statistical Methods section.

**Outcomes.** The primary outcome will be postoperative chest tube blood loss within 12 hours from the end of surgery. Chest tube blood loss is easy to measure, and a minimally subjective endpoint is unlikely to cause bias. The reason for not including intraoperative blood loss as a primary endpoint is that it is often difficult to accurately measure blood loss during surgery.

The secondary outcomes are as follows:

1) RBC transfusion volume during surgery and for 12 hours from the end of surgery

2) Prevalence of re-thoracotomy within 48 hours from the end of surgery

3) Prevalence of postoperative chest tube blood loss of ≥ 1000 mL within 12 hours from the end of surgery.

The safety outcomes are as follows:

1) Prevalence of any of the following infections: surgical site infection, mediastinitis, urinary tract infection, respiratory infection, central nervous system infection, gastrointestinal system infection, skin and soft tissue infection, or other infection

2) Prevalence of gross hematuria upon ICU admission

3) Prevalence of acute kidney injury within 7 days after surgery, defined in accordance with the Kidney Disease: Improving Global Outcomes criteria [17].

Safety outcomes 2) and 3) are set as indicators of mechanical hemolysis, one of the potential adverse events associated with salvaged RBC transfusion.

The exploratory outcomes are as follows:

1) Results of blood viscoelastic testing, prothrombin time, activated partial thromboplastin time, and platelet count

2) Transfusion volume during surgery and for 12 hours from the end of surgery (FFP without cryoprecipitate/cryoprecipitate/platelet concentrate)

3) Duration of surgery (hours)

4) Duration of CPB (hours)

5) Duration of postoperative mechanical ventilation (hours)

6) Duration of ICU stay (hours)

7) Mortality (all-cause death and cardiovascular death)

8) Heparin concentration of salvaged RBCs after processing remaining blood in the CPB circuit in the Cell Saver Elite+

9) Amount of protamine sulfate administered in the operating room and within 12 hours from the end of surgery.

**Participant timeline.** An unblinded researcher (AT or KY) will screen potential participants from the list of scheduled operations (until at least the day before the surgery). All screened patients, including all excluded and enrolled patients,

will be documented. From the recruited patients, written informed consent will be obtained by unblinded researchers (AT or KY) until the day of surgery and complete provisional registration. During CPB, eligibility for final registration will be assessed, and if met, an unblinded researcher (AT, KY or TM) will perform the allocation for the intervention until weaning from CPB. All outcomes will be assessed and data collected according to the participant timeline (Fig 1).

**Sample size estimation.** According to our institutional data, the mean and standard deviation of postoperative chest tube blood loss within 12 hours from the end of surgery in patients who met the same eligibility criteria as those in this randomized controlled trial and who received only allogeneic RBC transfusions were 674 mL and 400 mL, respectively. This corresponds to Class 1 bleeding (601–800 mL) as defined by the universal definition of perioperative bleeding proposed by Dyke et al. [18]. The universal definition categorizes bleeding severity in approximately 200-mL increments: the threshold between Class 0 (insignificant) and Class 1 (mild) is 600 mL; Class 2 (moderate) begins at 801 mL, and Class 3 (severe) begins at 1001 mL. On the basis of these clinically meaningful thresholds, we conservatively selected 200 mL as the noninferiority margin. With a one-sided α level of 0.025 and a 10% dropout rate, a sample size estimate of 142 randomized participants (71 for each group) has 80% power to demonstrate noninferiority of the salvaged RBC group to the allogeneic RBC group.

| | STUDY PERIOD | | | | | | |
| | Provisional Enrollment | Registration / Allocation | Post-allocation | | | | Close-out |
| TIMEPOINT* | 1 | 2 | 3 | 4 | 5 | 6 | 7 |
|---|---|---|---|---|---|---|---|
| **ENROLLMENT:** | | | | | | | |
| Eligibility screen | X | X | | | | | |
| Informed consent | X | | | | | | |
| Allocation | | X | | | | | |
| **INTERVENTIONS:** | | | | | | | |
| Salvaged RBC | | | X | | | | |
| Allogeneic RBC | | | X | | | | |
| **ASSESSMENTS:** | | | | | | | |
| Baseline variables | X | | | | | | |
| Laboratory tests | X | X | X | X | | | |
| Transfusion volume | | | ←——————→ | | | | |
| Chest tube blood loss | | | | ←———→ | | | |
| Re-thoracotomy | | | | ←————————→ | | | |
| Infections | | | | ←———————————————→ | | | |
| AEs and SAEs | | | | ←———————————————————→ | | | |

**Fig 1. SPIRIT schedule.** Timepoint 1: 1 day before surgery; Timepoint 2: during CPB; Timepoint 3: after weaning from CPB; Timepoint 4: at admission to the intensive care unit; Timepoint 5: 12 hours from the end of surgery; Timepoint 6: 48 hours from the end of surgery; and Timepoint 7: 30 days after surgery. Baseline variables comprise age, sex, height, weight, diagnosis, surgical procedure, medical history, medications, and laboratory values. AE adverse event, SAE severe adverse event.

**Recruitment.** Participant eligibility will be assessed by the anesthesiologist (AT) from the list of surgical applications. During the preoperative consultation, the operating surgeon will explain the procedure to the patient, obtain informed consent, and introduce the ongoing trial. Following this, the anesthesiologist will provide the patient with detailed information about the study.

Approximately 10 adult cardiovascular surgeries with CPB are performed each week in our institution. Of these, it is estimated that approximately two cases will meet the eligibility criteria. The enrollment period is expected to last approximately 18 months. No financial or nonfinancial incentives will be offered to trial investigators or participants to encourage enrollment.

## Assignment of interventions

**Allocation.** The principal investigator will enroll the participants. Participants will be randomly assigned to either the salvaged RBC transfusion group or the allogeneic RBC transfusion group with a 1:1 allocation using a computer-generated randomization schedule stratified by age (40–64 years or ≥ 65 years) and the presence or absence of aortic arch replacement with selective cerebral perfusion, using a permuted block design with random blocks. The block sizes will not be disclosed to ensure concealment. The principal investigator will access the Mujinwari system (Iruka System Corporation, Tokyo, Japan) via access to the internet and enter information about the selected stratification factors, which will be followed by allocation. Allocation concealment will be ensured because the principal investigator cannot predict the allocation sequence because of random block size. Additionally, the service will not release the randomization code until the patient has been recruited into the trial after all baseline measurements have been completed.

**Blinding (masking).** When obtaining written informed consent, participants will not be informed of the arm to which they will be assigned. Unblinded researchers will perform screening, registration, and allocation. The anesthesiologists and attending anesthesiologists will be told about the allocation. Drapes will be used to cover the infusion stand on which the infusion bags will be hung, making it impossible for anyone other than the anesthesiologist to see what is being administered. The surgeon, surgical assistants, postoperative care providers, and primary outcome assessor (ICU nurse) will be blinded to the treatment allocation during the trial procedure. Blinded staff will be allowed to consult the principal investigator when the blinded staff determines that unblinding is necessary owing to uncontrolled bleeding.

## Data collection, management, and analysis

**Data collection method.** A blinded ICU nurse will assess the primary outcome (postoperative chest tube blood loss). The ICU nurse will examine the bottle connected to the chest tube each hour and record the amount of blood loss in the electronic medical record. This recording is performed in daily practice; therefore, it does not require additional training, is less susceptible to measurement bias, and yields an outcome with minimal subjectivity. The researcher will refer to the electronic medical record, total the blood loss from all chest tubes, and record the amount of blood loss recorded by the ICU nurse for 12 hours in the case report form (CRF). If the participant enters the ICU at x hours and 0–29 minutes after surgery, the primary outcome will be recorded from ICU admission to x + 12 hours. If the participant enters the ICU at x hours and 30–59 minutes, the primary outcome will be recorded from ICU admission to x + 13 hours. If 12 hours have passed since the end of the initial surgery during re-thoracotomy, the amount of blood loss recorded in the operating room to that point will be added to the postoperative chest tube blood loss as the primary outcome. Other data, namely baseline characteristics and secondary outcomes, will be collected from the electronic database. Among these, the total RBC transfusion volume will be recorded for all patients, and in the intervention group, the transfused RBC volume will additionally be documented separately as cell-salvaged and allogeneic sources. All data will be recorded in the respective CRF by unblinded researchers.

**Data management.**  To ensure the correctness and completeness of the data, a person in charge of monitoring will perform plausibility checks of the CRF. Inconsistencies in the data will be queried. Responses to queries will be documented directly in the monitoring report. By signing the CRF, the principal investigator will confirm that all procedures meet the protocol requirements and that complete and reliable data have been entered into the CRF. Two researchers will independently enter the data into the database, and consistency between the two entries will be tested. The database will be created using Microsoft Excel 2019 (Microsoft Corp., Redmond, WA, USA).

**Statistical methods.**  The modified intention-to-treat (mITT) set will include all participants who will be randomized according to randomized treatment assignment and whose outcome data will be obtained. The per-protocol set is defined as a subset of the mITT population from which participants with major protocol violations will be excluded. These include violations of eligibility criteria, improper administration of salvaged or allogeneic RBCs, or marked violations in preoperative, concomitant, or postoperative care that conflict with the study protocol. The safety analysis set is defined as all randomized participants who start transfusion with salvaged or allogeneic RBCs. The primary and secondary outcomes will be analyzed primarily for the mITT set and repeated, for sensitivity reasons, for the per-protocol set. The safety outcomes and adverse events will be analyzed for the safety analysis set.

Noninferiority of salvaged to allogeneic RBC transfusions will be declared if the upper limit of the confidence interval (CI) (for the mean difference in postoperative chest tube blood loss [value in the salvaged RBC group minus the value in the allogeneic RBC group]) is < 200 mL. The following subgroup analyses of the primary outcome will be performed: 1) age between 40 and 64 years versus ≥ 65 years, 2) with/without cardiac surgery history, 3) with/without major vascular surgery with selective cerebral perfusion, and 4) total volume of intraoperative salvaged and allogeneic RBC transfusions after weaning from CPB ≥ 800 mL versus < 800 mL.

In the secondary outcomes, RBC transfusion volume will be the test of superiority, and the incidence of re-thoracotomy and > 1000 mL bleeding within 12 hours after ICU admission will be tests of noninferiority of salvaged RBC transfusion versus allogeneic RBC transfusion. Noninferiority of salvaged to allogeneic RBC transfusions will be declared if the upper limit of the CI for the difference in the incidence of re-thoracotomy and >1000 mL bleeding within 12 hours after ICU admission is less than 2% and 3%, respectively.

According to the safety outcomes, the obtained data will be summarized for the entire population and for each allocation group using appropriate statistical measures; i.e., for continuous variables, as the mean and standard deviation or median and quantiles, and for categorical variables, as frequencies (number of cases and participants) and proportions. Two-sided 95% CIs will also be provided.

In all statistical analyses, the significance of the outcome will be defined as a one-tailed p-value <0.025 for the noninferiority tests or a two-tailed p-value <0.05 for the superiority tests. Interim analyses have not been planned. There will be no criteria for early discontinuation of the study. Because the primary and secondary outcomes data are collected as part of standard clinical practice, these data are rarely missing; therefore, these analyses will be performed by excluding any missing values. If more than 5% of the data for a given outcome are missing, we will assess the pattern and mechanism of missingness. Depending on the findings, appropriate sensitivity analyses will be performed, such as multiple imputation under the missing at random assumption, to evaluate the robustness of the results. All statistical analyses will be performed using R software (R Foundation for Statistical Computing; www.r-project.org).

## Monitoring

**Monitoring and auditing.**  In this study, staff designated by the principal investigator will be in charge of quality control of the study and will perform monitoring to ensure, enhance, and protect participants' safety, well-being, and rights, and to check the scientific quality of the clinical research and the reliability of the data. The monitoring staff will comprise coinvestigators in the Department of Anesthesiology who do not monitor the work for which they are directly responsible. The monitoring staff are independent of the sponsors, with no competing interests, and an audit is not planned.

**Safety considerations.** All adverse events, including serious adverse events, will be recorded and reported by the principal investigator or coinvestigators. These events will be documented in the CRF and assessed in accordance with the definitions and procedures outlined in the S2 and S3 Files. Each event will be evaluated to determine whether it is expected or unexpected, and whether a causal relationship to the study intervention is suspected.

To ensure protocol compliance and participant safety, monitoring will be performed by designated coinvestigators in the Department of Anesthesiology who are not involved in the care of their own participants. These monitors are independent of the study sponsor and have no competing interests. They will regularly review the CRFs and communicate with study personnel to facilitate timely identification and management of adverse events and protocol violations. Although an external audit is not planned because of budget limitations, internal audits will be performed periodically by the monitoring staff.

## Ethics and dissemination

**Ethics approval and consent to participate.** The study protocol was approved by the National Cerebral Cardiovascular Center Research Ethics Committee, Suita, Japan on 25 July 2024 (approval number: R24035). Investigators in the Department of Anesthesiology will obtain written informed consent from all patients before enrollment. The voluntary willingness of all participants will be ascertained after providing comprehensive written and verbal explanations regarding the study.

**Protocol amendments.** If the research protocol needs to be revised, the principal investigator will discuss this with other investigators. The principal investigator will then submit the revised protocol to the ethics board and chairperson of the research institution. After the institutional ethics board reviews and approves the revision, permission will be obtained from the chairperson of the research institution.

**Confidentiality.** At enrollment, all participants will be identified by individual registry numbers to anonymize the CRFs and other documents. This individual registry number will consist of numbers unrelated to the information that can identify a specific individual. The principal investigator will create and maintain the list linking the anonymized data to each participant. Under the supervision of the principal investigator, all data and documents will be secured from unauthorized access in locked cabinets with restricted access. Data analysis will be performed using anonymized data.

**Access to data.** During the trial, access to the final trial dataset will be restricted to the principal investigator (AT) and the trial statistician (SO). There are no contractual agreements that limit their access to the data. Upon completion of the trial and following appropriate anonymization, the full deidentified participant-level dataset will be deposited in the Dryad digital repository, where the dataset will be publicly accessible to facilitate verification, replication, and secondary analyses by other researchers.

**Dissemination policy.** The study results will be presented in appropriate international scientific journals and at scientific conferences. There are no contractual restrictions on publication. A professional writing service will not be engaged; the manuscript will be prepared by the investigators.

**Status and timeline of the study.** Recruitment of participants in this study began on 22 August 2024 and is ongoing. We estimate that recruitment will end in September 2026. Data collection is expected to be completed by March 2027, and the study results are anticipated to be available by March 2028. The initial protocol (Version 1.1) was finalized on 11 July 2024. The protocol was amended to Version 1.2 on 13 September 2024 to broaden the eligibility criteria because of a disappointing recruitment rate and to remove the intraoperative platelet transfusion criterion. Version 1.3, finalized on 4 December 2024, included further relaxation of the eligibility criteria and modifications to the secondary and exploratory outcomes. The protocol currently in use is Version 1.3.

## Discussion

The use of intraoperative cell salvage during cardiovascular surgery is strongly recommended but is primarily limited to procedures with a low risk of bleeding [5,6]. However, to our knowledge, no high-quality randomized controlled trials have

been performed to investigate whether intraoperative RBC salvage causes coagulopathy in procedures with a high risk of bleeding. This study aims to address this gap, with high internal validity. If intraoperative RBC salvage proves noninferior to allogeneic RBC blood transfusion regarding postoperative blood loss, the use of salvaged RBC transfusions could be promoted in patients undergoing high-risk procedures. This, in turn, could lead to a reduction in the use of allogeneic RBC transfusions, a decrease in transfusion-related adverse events, and substantial healthcare cost savings.

This study has several limitations. First, this study will not be double-blinded; the anesthesiologist will be aware of the allocation. At the facility where the study will be performed, the anesthesiologist will independently determine the type, timing, and volume of blood products to administer during surgery. As a result, the lack of blinding may introduce differences in transfusion therapy between the groups. To address this, we will standardize the criteria for transfusion administration and hemostasis-related treatments, including the use of tranexamic acid, to ensure consistency between the groups. Furthermore, recognizing this potential bias, we deliberately selected an objective endpoint—postoperative chest tube blood loss within 12 hours after surgery—as the primary outcome. This parameter reflects actual bleeding volume and is less susceptible to performance bias from clinical decision-making, including that of the anesthesiologist. We believe this choice strengthens the internal validity of the trial. Second, this study will focus on patients with a high bleeding risk; however, patients with an extremely high bleeding risk will be excluded. This exclusion is because including such patients would increase the standard deviation of the primary outcome, bleeding volume, which would require a larger sample size and reduce the feasibility of the study. Therefore, extrapolation of the results of this study to patients with an extremely high bleeding risk is limited. Third, although the study design defines high bleeding risk on the basis of age and type of procedure, participants with low actual intraoperative bleeding may still be enrolled if they develop hemodilution-induced anemia during CPB. This situation is more likely in participants with preoperative anemia or small body size. Such participants will not be excluded because their exclusion could markedly reduce the number of eligible participants. To address this limitation, we plan to perform subgroup analyses on the basis of postoperative bleeding volume after CPB weaning, which may serve as an indirect indicator of actual transfusion needs. Fourth, the hemoglobin concentration of salvaged RBCs is comparable to or slightly lower than that of allogeneic RBCs [8–10], potentially increasing the risk of dilutional coagulopathy in the salvaged RBCs group. However, because the difference in hemoglobin concentration between the two types of RBC preparations is small, we believe that the clinical impact is likely to be limited. Importantly, because it would likely increase bleeding in the salvaged RBCs group, such bias would reduce the likelihood of demonstrating noninferiority, rather than lead to a false-positive result.

## Supporting information

**S1 File. SPIRIT checklist.**
(PDF)

**S2 File. Study protocol in English.**
(DOCX)

**S3 File. Study protocol in Japanese.**
(DOCX)

## Acknowledgments

We thank Jane Charbonneau, DVM, from Edanz (https://jp.edanz.com/ac) for editing a draft of this manuscript.

## Author contributions

**Conceptualization:** Akito Tsukinaga, Kenji Yoshitani, Shin Ito, Hiroko Kanazawa, Akira Shimokawa, Sho Masuda, Yoshiaki Ito, Shinnosuke Miura, Naonori Kawamoto, Yosuke Inoue, Satsuki Fukushima, Hitoshi Matsuda, Takuma Maeda.

**Formal analysis:** Soshiro Ogata.

**Funding acquisition:** Kenji Yoshitani, Takuma Maeda.

**Methodology:** Akito Tsukinaga, Soshiro Ogata, Takahiro Mihara, Shin Ito, Sho Masuda.

**Project administration:** Akito Tsukinaga, Kenji Yoshitani.

**Resources:** Akito Tsukinaga.

**Writing – original draft:** Akito Tsukinaga.

**Writing – review & editing:** Kenji Yoshitani, Soshiro Ogata, Takahiro Mihara, Shin Ito, Hiroko Kanazawa, Akira Shimokawa, Sho Masuda, Masahiro Morinaga, Yoshiaki Ito, Shinnosuke Miura, Naonori Kawamoto, Yosuke Inoue, Satsuki Fukushima, Hitoshi Matsuda, Takuma Maeda.

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
