## [Decision Letter · Decision Letter 0]

15 Jul 2025

Dear Dr. Tsukinaga,

We look forward to receiving your revised manuscript.

Kind regards,

Mehmet Baysal

Academic Editor

PLOS ONE

Journal Requirements: 

2.Your ethics statement should only appear in the Methods section of your manuscript. If your ethics statement is written in any section besides the Methods, please move it to the Methods section and delete it from any other section. Please ensure that your ethics statement is included in your manuscript, as the ethics statement entered into the online submission form will not be published alongside your manuscript.

Reviewers' comments:

Reviewer's Responses to Questions

**Comments to the Author**

1. Does the manuscript provide a valid rationale for the proposed study, with clearly identified and justified research questions?

Reviewer #1: Yes

Reviewer #2: Yes

Reviewer #3: Partly

Reviewer #4: Yes

Reviewer #5: Yes

2. Is the protocol technically sound and planned in a manner that will lead to a meaningful outcome and allow testing the stated hypotheses?

Reviewer #1: Partly

Reviewer #2: Yes

Reviewer #3: Partly

Reviewer #4: Yes

Reviewer #5: Yes

3. Is the methodology feasible and described in sufficient detail to allow the work to be replicable?

Reviewer #1: Yes

Reviewer #2: Yes

Reviewer #3: Yes

Reviewer #4: Yes

Reviewer #5: Yes

4. Have the authors described where all data underlying the findings will be made available when the study is complete?

Reviewer #1: No

Reviewer #2: No

Reviewer #3: No

Reviewer #4: No

Reviewer #5: No

5. Is the manuscript presented in an intelligible fashion and written in standard English?

Reviewer #1: Yes

Reviewer #2: Yes

Reviewer #3: Yes

Reviewer #4: Yes

Reviewer #5: Yes

You may also provide optional suggestions and comments to authors that they might find helpful in planning their study.

Reviewer #1: I have read with interest the manuscript submitted describing the protocol of a new RCT on the cell-salvaged vs. allogeneic red blood cells transfusion in cardiac surgery at high risk of bleeding. I have a few comments and observation about the paper.

Title: I don't believe "safety" correctly describes the final aim of the paper. As the authors state, this is a non-inferiority trial with blood loss as primary endpoint. The safety outcomes are mentioned within the manuscript, after the primary and the secondary outcomes, and only the "prevalence of any infection" has to do with actual safety. Indeed, no other Severe Adverse Events that might be evaluated as safety among the two groups are described. The title should be better formulated to reflect this aspect.

Exclusion criteria: Please define "extremely high risk of bleeding".

Please add a paragraph about the management of anemic patients that need to be transfused during CPB (are they transfused preoperatively or perioperatively? are they excluded or maintained in the study).

Is there a bleeding management algorithm including point-of-care testing in the Institution?

How FFP or Platelet Concentrate transfusion managed if it is required immediately after CPB weaning (because anticipated due to patient's preoperative paramenters or to uncontrolled bleeding after CPB weaning).

Blood viscosity testing is mentioned among the Safety outcomes. Please give more technical details of how and when this testing is performed, the parameters that will be considered abnormal/at risk, and how this affects safety of the patient.

Reviewer #2: Authors plan to enroll 142 patients with age at least 40 into a randomized, two-arm (salvaged RBC or allogeneic RBC transfusions intraoperatively) clinical trial to evaluate the safety of salvaged RBC transfusion in patients with a high bleeding risk. The authors consider12-hour postoperative chest tube blood loss as the primary outcome and also several secondary outcomes.

1. Please provide justification for the non-inferiority margin of 200 ml.

2. Authors plan to exclude the patients with extremely high bleeding risk. Please define or how to identify “extremely high risk” here.

3. It would be great to lay out data access process.

4. Please comment on how to deal with missing data.

Reviewer #3: In the present manuscript, Tsukinaga et al describe a study protocol designed to compare the respective values of autologous versus allogenic transfusion in the setting of high risk cardiac surgery. The question is of interest and the results might be of interest to the reader. However the protocol itself has structural flaws, especially the consideration of the primary endpoint wich is irellevant to the question, and the manuscript presents no data/result. Therefore it does not represent a work worthy of publication in its current state.

Reviewer #4: I thought this was a very clear, well written protocol. My only concern is with regard to question 4. The authors state that the data will only be available to the principal investigator and statistician. They need to explain how the anonymized data will be publicly available. A secondary minor comment: pay attention to the use of capital letters in journal titles. All significant words should be capitalized.

Reviewer #5: The protocol is well-designed with clear objectives, methodology, appropriate control groups, statistics and ethical safeguards. It addresses a relevant clinical question and ensures reproducibility.

Providing clarification regarding the issues listed below could further strengthen the study:

1. Defining ‘High risk bleeding’ and ‘extremely high bleeding risk’ in the inclusion and exclusion criteria of the study population

2. The anaesthesiologist, who is aware of the allocation, also manages transfusions. This can result in RBC usage bias.

3. Rationale for taking 200 ml as cut off between the salvaged and allogenic transfused group.

4. The protocol suggests patients with major protocol violations will be excluded from PP analysis, but doesn’t explain how many deviations are expected or how these will be minimized.

5. Rationale for choosing liberal transfusion triggers while recent studies advocate restrictive transfusion protocols.

6. Volume of transfusion (salvaged blood vs allogenic blood) in both the study arm may differ and might affect the outcome.

7. Salvaged blood has a higher risk of hemolysis and infection. Specifying reporting and monitoring of adverse event method, agency and an audit strategy would promote transparency.

**Do you want your identity to be public for this peer review?** For information about this choice, including consent withdrawal, please see our Privacy Policy

Reviewer #1: No

Reviewer #2: No

Reviewer #3: No

Reviewer #4: **Yes: ** Mary Berg, MD

Reviewer #5: No

---

## [Author Response · Author response to Decision Letter 1]

23 Aug 2025

Manuscript Number: PONE-D-25-26936  

Intraoperative cell-salvaged versus allogeneic red blood cell transfusions in high-bleeding-risk cardiovascular surgery: Protocol for a single-center, randomized, parallel-group, non-inferiority trial

August 22, 2025

Dear Dr Baysal,

Attached is a revised version of our manuscript titled “Intraoperative cell-salvaged versus allogeneic red blood cell transfusions in high-bleeding-risk cardiovascular surgery: Protocol for a single-center, randomized, parallel-group, non-inferiority trial,” which we would like to resubmit for publication in PLOS ONE.

Your comments were highly insightful and enabled us to greatly improve the quality of our manuscript. We present our point-by-point responses to the comments on the following pages.

In accordance with your suggestions, we have revised some text for better clarity. We hope that our revisions and accompanying responses will be sufficient to make our manuscript suitable for publication in PLOS ONE.

We look forward to hearing from you at your earliest convenience.

Yours sincerely,

Akito Tsukinaga

Email: akito.tsukinaga30303@gmail.com

Responses to the Academic Editor

Comment: One of the main concerns with the manuscript is how to overcome bias and missing data. The authors should mitigate, and account for potential biases and missing data in their study to ensure robust, reliable, and generalizable findings.

Response:

1. Potential Bias:

We understand the Academic Editor’s concern regarding potential bias in the study, particularly related to intraoperative transfusion management. Because the attending anesthesiologist, who is responsible for transfusion decisions, cannot be blinded, there is a risk of performance bias in the amount of RBCs administered, even when hemoglobin thresholds are predefined. Recognizing this limitation, we deliberately avoided using RBC transfusion volume as the primary outcome. Instead, we selected postoperative chest tube blood loss within 12 hours from the end of surgery as the primary endpoint, as this is an objective and quantifiable measure that is less susceptible to clinician discretion. While RBC transfusion volume remains a secondary outcome and will be analyzed accordingly, we do not intend to draw definitive conclusions based solely on this measure. We believe that this outcome selection minimizes the risk of performance bias and strengthens the internal validity of the study.

We have added this rationale to the revised manuscript (Page 32, Lines 548–552 in clear version; Line 644–648 in tracked version).

2. Missing Data:

We recognize that missing data may introduce bias and reduce the reliability of the findings. Missing data in the primary and secondary outcomes are expected to be rare because these data are collected as part of routine clinical practice. Therefore, we will use complete case analysis as the primary approach. However, if the rate of missing data exceeds 5%, we will evaluate the mechanism of missingness and perform sensitivity analyses using appropriate imputation methods, such as multiple imputation. This approach aligns with best practices and enhances the robustness of our findings. The revised plan is now clearly stated in the manuscript (Page 27, Lines 458–462 in clear version; Line 540–544 in tracked version).

We believe these considerations will ensure the internal validity and reliability of our findings. Relevant revisions have been incorporated into the manuscript.

Responses to Reviewer #1

Point 1:

Title: I don't believe "safety" correctly describes the final aim of the paper. As the authors state, this is a non-inferiority trial with blood loss as primary endpoint. The safety outcomes are mentioned within the manuscript, after the primary and the secondary outcomes, and only the "prevalence of any infection" has to do with actual safety. Indeed, no other Severe Adverse Events that might be evaluated as safety among the two groups are described. The title should be better formulated to reflect this aspect.

Response:

We thank the reviewer for this valuable comment. We agree that the term “safety” in the title may not accurately reflect the study’s primary aim, which is to evaluate the noninferiority of salvaged red blood cell (RBC) transfusion compared with allogeneic RBC transfusion in terms of postoperative blood loss. In response, we revised not only the title but also the relevant wording in the abstract to ensure consistency and better reflect the objective of evaluating a clinically relevant outcome, rather than emphasizing safety (Page 1, Lines 1–3 and Page 3, Lines 35–39 in clear version; Line 44–48 in tracked version).

Point 2:

Exclusion criteria: Please define "extremely high risk of bleeding".

Response:

We appreciate this being pointed out. We agree that the term “extremely high risk of bleeding” requires a clear operational definition to enhance reproducibility and transparency. On the basis of our institutional experience and the clinical records of previously excluded cases, we define “extremely high risk of bleeding” as either a preoperative platelet count of < 100 × 10⁹/L, or ≥ 2 prior cardiovascular surgeries via median sternotomy.

We have added this definition in the exclusion criteria in the revised manuscript (Page 8, Lines 116–117 in clear version; Line 211–212 in tracked version). We believe this clarification improves reproducibility and transparency. Because we have provided a clear definition of “extremely high risk of bleeding,” the third limitation regarding the subjectivity of this criterion has been removed from the revised manuscript (Page 32, Line 557 in clear version; Line 725 in tracked version).

Point 3:

Please add a paragraph about the management of anemic patients that need to be transfused during CPB (are they transfused preoperatively or perioperatively? are they excluded or maintained in the study).

Response:

We thank the reviewer for this important question. We have clarified the perioperative management of anemic patients in the “Intervention – concomitant care” section (Page 12, Lines 195–197 in clear version; Line 292–294 in tracked version). Treatment for preoperative anemia, including RBC transfusion, is not defined by the study protocol; rather, the decision regarding this therapy is left to the discretion of the attending physician based on clinical judgment, such as symptoms of anemia or underlying comorbidities. Because this decision will be made before final registration and randomization, it is unlikely to introduce bias into the trial.

If an anemic patient proceeds to surgery without a preoperative transfusion, hemodilution during CPB may reduce hemoglobin levels below the predefined threshold of 8 g/dL, prompting intraoperative transfusion and qualifying the patient for final registration. We acknowledge that this design may allow enrollment of patients with low actual intraoperative bleeding risk—particularly those with preoperative anemia or small body size—if they require transfusion primarily because of hemodilution rather than bleeding. Such patients were not excluded because their exclusion could have markedly reduced the number of eligible participants and compromised study feasibility. To address this limitation, we plan to perform subgroup analyses on the basis of postoperative bleeding volume after CPB weaning, which may serve as an indirect indicator of actual transfusion requirements. This information has been added to the “Limitations” section of the revised manuscript (Page 32–33, Line 557–565 in clear version; Line 725–758 in tracked version).

Point 4:

Is there a bleeding management algorithm including point-of-care testing in the Institution?

Response:

We thank the reviewer for this important question. Although our institution has access to point-of-care coagulation testing devices, such as Teg® 6s (Haemonetics, Boston, MA, USA), Quantra® Hemostasis Analyzer (HemoSonics, Charlottesville, VA, USA), and Fibcare® (Atom Medical, Kawasaki, Japan), there is no standardized bleeding management algorithm based on these platforms. Instead, clinical decisions during surgery are made at the discretion of the attending anesthesiologist. To minimize inter-physician variability and ensure consistent perioperative management during the study, we have clarified and standardized our transfusion practices by establishing a protocol-based strategy (Page 13, Line 221 and Page 14, Lines 223–236 in clear version; Page 13, Line 320 and Page 14, Lines 336–349 in tracked version).

Point 5:

How FFP or Platelet Concentrate transfusion managed if it is required immediately after CPB weaning (because anticipated due to patient's preoperative paramenters or to uncontrolled bleeding after CPB weaning).

Response:

We thank the reviewer for this valuable question. We have provided a detailed description of the management of FFP and platelet concentrate transfusions in the revised manuscript (Page 13, Line 221 and Page 14, Lines 223–236 in clear version; Page 13, Line 320 and Page 14, Lines 336–349 in tracked version).

Additionally, to guide transfusion management and assess coagulation status, we will perform not only standard laboratory tests but also point-of-care testing, such as blood gas analysis and ACT, with viscoelastic testing at specific perioperative time points. These details are described in the newly added “Intervention – Laboratory and Viscoelastic Testing” section (Page 15–17, Line 249–282 in clear version; Line 386–402 in tracked version).

Point 6:

Blood viscosity testing is mentioned among the Safety outcomes. Please give more technical details of how and when this testing is performed, the parameters that will be considered abnormal/at risk, and how this affects safety of the patient.

Response:

We thank the reviewer for making this insightful comment. Based on the feedback, we have revised the protocol to reclassify viscoelastic testing from a safety outcome to an exploratory outcome (Page 19, Line 316–319 in clear version; Line 445–448 in tracked version). This change was made because the primary role of viscoelastic testing in this study is to provide supportive, hypothesis-generating information rather than to directly assess patient safety. If noninferiority of the primary endpoint (postoperative chest tube blood loss) is not demonstrated, analysis of these exploratory parameters may help identify potential mechanisms, such as residual heparin effect (elevated clot time ratio), thrombocytopenia or platelet dysfunction (reduced platelet contribution to clot stiffness), or hypofibrinogenemia/fibrinogen dysfunction (reduced fibrinogen contribution to clot stiffness).

In response, we have added related technical details in a newly created subsection titled “Intervention – Laboratory and Viscoelastic Testing” (Page 16–17, Line 269–282 in clear version; Line 389–402 in tracked version) in the revised manuscript. This section describes how and when the testing will be performed, which parameters will be considered abnormal or indicative of risk, and how these findings will relate to patient safety.

Responses to Reviewer #2

Point 1:

Please provide justification for the non-inferiority margin of 200 ml.

Response:

We thank the reviewer for this comment. The noninferiority margin of 200 mL was determined on the basis of our institutional data and previous studies. According to our preliminary data, patients who met the eligibility criteria for this trial and received only allogeneic RBC transfusions had a mean postoperative chest tube blood loss of 674 mL within 12 hours from the end of surgery. This corresponds to Class 1 bleeding (601–800 mL) as defined by the Universal Definition of Perioperative Bleeding (UDPB) proposed by Mehta et al. (Ann Thorac Surg. 2009;87:148–156). The UDPB categorizes bleeding severity in approximately 200-mL increments: the threshold between Class 0 (insignificant) and Class 1 (mild) is 600 mL; Class 2 (moderate) begins at 801 mL, and Class 3 (severe) begins at 1001 mL. On the basis of these clinically meaningful thresholds, we conservatively selected 200 mL as the noninferiority margin.

We have added this rationale to the main manuscript (Page 21, lines 349–355 in clear version; Line 481–487 in tracked version).

Point 2:

Authors plan to exclude the patients with extremely high bleeding risk. Please define or how to identify “extremely high risk” here.

Response:

We thank the reviewer for this helpful comment. We agree that the term “extremely high risk of bleeding” requires a clear operational definition to enhance reproducibility and transparency. On the basis of our institutional experience and clinical records of previously excluded cases, we define “extremely high risk of bleeding” as either a preoperative platelet count of < 100 × 10⁹/L, or ≥ 2 prior cardiovascular surgeries via median sternotomy. We have added this definition to the exclusion criteria in the revised manuscript (Page 8, Lines 116–117 in clear version; Line 211–212 in tracked version). We believe this clarification improves reproducibility and transparency. Because we have now provided a clear definition of “extremely high risk of bleeding,” the third limitation regarding the subjectivity of this criterion has been removed from the revised manuscript (Page 32, Line 557 in clear version; Line 725 in tracked version).

Point 3:

It would be great to lay out data access process.

Response:

We appreciate this valuable suggestion. In response, we have revised the manuscript to clearly describe the data access process in accordance with the SPIRIT 2013 guidelines. Specifically, we have updated the “Access to data” section (Page 30, Line 509–514 in clear version; Line 673–678 in tracked version) to clarify data access during the trial. Furthermore, we have revised the “Data Availability Statement” section (Page 34, Lines 586–589 in clear version; Line 769–772 in tracked version) to specify that, upon completion of the trial, the deidentified dataset will be deposited in a publicly accessible repository (i.e., Dryad) to facilitate verification, replication, and secondary analyses by other researchers.

Point 4:

Please comment on how to deal with missing data.

Response:

We appreciate the comment regarding missing data. We recognize that missing data may introduce bias and reduce the reliability of the findings. Missing data in the primary and secondary outcomes are expected to be rare because these data are collected as part of routine clinical practice. Therefore, we will use complete case analysis as the primary approach. However, if the rate of missing data exceeds 5%, we will evaluate the mechanism of missingness and perform sensitivity analyses using appropriate imputation methods, such as multiple imputation. This approach aligns with best practices and enhances the robustness of our findings. The revised plan is now clearly stated in the manuscript (Page 27, Lines 458–462 in clear version; Line 540–544 in tracked version).

Responses to Reviewer #3

Point: In the present manuscript, Tsukinaga et al describe a study protocol designed to compare the respective values of autologous versus allogeneic transfusion in the setting of high risk cardiac surgery. The question is of interest and the results might be of interest to the reader. However the protocol itself has structural flaws, especially the consideration of the primary endpoint wich is irellevant to the question, and the manuscript presents no data/result. Therefore it does not represent a work worthy of publication in its current state.

Response:

We appreciate the thoughtful review and feedback. We respectfully disagree with the assertion that our protocol has structural flaws or that the primary endpoint is irrelevant to the study question.

Our primary objective is to assess whether intraoperative transfusion of cell-salvaged red blood cells (RBCs) is noninferior to allogeneic RBCs in terms of perioperative bleeding risk. As such, we selected postoperative chest tube blood loss within 12 hours from the end of surgery as the primary endpoint, which is a direct and objective measure of early postoperative bleeding. This endpoint is clinically relevant and has been used in previous transfusion-related studies

---

## [Decision Letter · Decision Letter 1]

15 Sep 2025

Dear Dr. Tsukinaga,

We look forward to receiving your revised manuscript.

Kind regards,

Mehmet Baysal

Academic Editor

PLOS ONE

Journal Requirements:

Reviewers' comments:

Reviewer's Responses to Questions

**Comments to the Author**

1. Does the manuscript provide a valid rationale for the proposed study, with clearly identified and justified research questions?

Reviewer #2: Yes

Reviewer #3: Partly

Reviewer #4: Yes

Reviewer #5: Yes

2. Is the protocol technically sound and planned in a manner that will lead to a meaningful outcome and allow testing the stated hypotheses?

Reviewer #2: Yes

Reviewer #3: Partly

Reviewer #4: Yes

Reviewer #5: Yes

3. Is the methodology feasible and described in sufficient detail to allow the work to be replicable?

Reviewer #2: Yes

Reviewer #3: Yes

Reviewer #4: Yes

Reviewer #5: Yes

4. Have the authors described where all data underlying the findings will be made available when the study is complete?

Reviewer #2: Yes

Reviewer #3: Yes

Reviewer #4: Yes

Reviewer #5: Yes

5. Is the manuscript presented in an intelligible fashion and written in standard English?

Reviewer #2: Yes

Reviewer #3: Yes

Reviewer #4: Yes

Reviewer #5: Yes

You may also provide optional suggestions and comments to authors that they might find helpful in planning their study.

Reviewer #2: Thanks for appropriately addressing all the comments. This reviewer has no further concerns on this revised manuscript.

Reviewer #3: While the authors have addressed most of the reviewers’ comments, some of their responses introduce additional concerns. For example, the definition of “extremely high risk of bleeding” combines a reasonable criterion (two or more prior surgeries) with another that is less appropriate (platelet count below 100). The latter does not in itself confer an extremely high bleeding risk and can be readily managed with preoperative platelet transfusion.

More generally, the authors—likely surgeons and anesthesiologists—would benefit from broadening the interdisciplinary basis of their research and involving colleagues such as transfusion medicine specialists and hematologists in order to refine their study protocol and ensure its relevance to clinical practice.

Reviewer #4: Thank you for the opportunity to again review this protocol. The edits made by the authors have clarified the questions asked after the initial review. This well-designed study will be of great interest to anyone who is interested in the optimal use of perioperative blood salvage.

Just a point of clarification, on page 10, lines 161 -163: When you say that you will set up the cell saver, will that be done for all cases, whether or not it will be used? For the sake of preserving the blinding, it would need to be set up for even the control cases and a sham procedure (i.e. running of the instrument) should be done when salvaged blood will not be processed or transfused so that the blinded participants in the room will not know if it is actually being used or not.

Another point of clarification, on page 18, line 309, the first secondary outcome: This might be more of an exploratory outcome. For the study group, will you record how much of the RBC volume transfused is from cell salvage vs. banked RBCs? It would be interesting to see if there is a tendency to see more bleeding in cases in which a larger proportion of RBCs transfused comes from salvaged blood.

Reviewer #5: All suggestions have been considered.I believe the manuscript has improved in clarity and scientific value.

**Do you want your identity to be public for this peer review?** For information about this choice, including consent withdrawal, please see our Privacy Policy

Reviewer #2: No

Reviewer #3: No

Reviewer #4: No

Reviewer #5: No

---

## [Author Response · Author response to Decision Letter 2]

23 Sep 2025

Manuscript Number: PONE-D-25-26936  

Intraoperative cell-salvaged versus allogeneic red blood cell transfusions in high-bleeding-risk cardiovascular surgery: Protocol for a single-center, randomized, parallel-group, non-inferiority trial

September 23, 2025

Dear Dr Baysal,

Attached is a revised version of our manuscript titled “Intraoperative cell-salvaged versus allogeneic red blood cell transfusions in high-bleeding-risk cardiovascular surgery: Protocol for a single-center, randomized, parallel-group, non-inferiority trial,” which we would like to resubmit for publication in PLOS ONE.

Your comments were highly insightful and enabled us to greatly improve the quality of our manuscript. We present our point-by-point responses to the comments on the following pages.

In accordance with your suggestions, we have revised some text for better clarity. We hope that our revisions and accompanying responses will be sufficient to make our manuscript suitable for publication in PLOS ONE.

We look forward to hearing from you at your earliest convenience.

Yours sincerely,

Akito Tsukinaga

Email: akito.tsukinaga30303@gmail.com

Responses to Reviewer #2

Comment: Thanks for appropriately addressing all the comments. This reviewer has no further concerns on this revised manuscript.

Response:

We sincerely thank the reviewer for their positive evaluation and are grateful for their supportive comments.

Responses to Reviewer #3

Point 1: While the authors have addressed most of the reviewers’ comments, some of their responses introduce additional concerns. For example, the definition of “extremely high risk of bleeding” combines a reasonable criterion (two or more prior surgeries) with another that is less appropriate (platelet count below 100). The latter does not in itself confer an extremely high bleeding risk and can be readily managed with preoperative platelet transfusion.

Response: We appreciate the reviewer’s insightful comment. In response, we have removed the platelet count <100 × 10⁹/L criterion and now define “extremely high risk of bleeding” solely as patients with two or more prior sternotomies. The initial rationale for including the platelet count threshold was to exclude patients with coagulopathy, such as chronic disseminated intravascular coagulation (DIC) associated with aortic aneurysm, since these patients may still exhibit a bleeding tendency intraoperatively even if preoperative platelet transfusion normalizes the platelet count. However, as the reviewer correctly noted, not all patients with platelet counts below 100 × 10⁹/L necessarily present with coagulopathy. At the same time, it is difficult to establish a clear and objective definition to specifically exclude cases of coagulopathy associated with aortic aneurysm. Therefore, in the revised manuscript, we have clarified that for patients considered to be at extremely high risk of bleeding, the final decision on exclusion will be made through discussion between the surgeon and the anesthesiologist.

We have revised the Exclusion criteria section accordingly, which now reads as follows:

“Extremely high risk of bleeding, defined as patients with two or more prior cardiovascular surgeries via median sternotomy. For other cases in which the bleeding risk is considered to be extremely high, the final decision on exclusion will be made through discussion between the surgeon and the anesthesiologist.” (Page 8, Lines 116–119 in clean version)

Point 2: More generally, the authors—likely surgeons and anesthesiologists—would benefit from broadening the interdisciplinary basis of their research and involving colleagues such as transfusion medicine specialists and hematologists in order to refine their study protocol and ensure its relevance to clinical practice.

Response: We thank the reviewer for this thoughtful suggestion. We would like to clarify that the first author (AT) and co-author (KY) are anesthesiologists with expertise in transfusion medicine, as they are affiliated with the Department of Transfusion. We have drawn upon this expertise in planning the present study protocol.

Responses to Reviewer #4:

Comment: Thank you for the opportunity to again review this protocol. The edits made by the authors have clarified the questions asked after the initial review. This well-designed study will be of great interest to anyone who is interested in the optimal use of perioperative blood salvage.

Response: We sincerely thank the reviewer for the positive evaluation and encouraging comments regarding our study.

Point 1: Just a point of clarification, on page 10, lines 161 -163: When you say that you will set up the cell saver, will that be done for all cases, whether or not it will be used? For the sake of preserving the blinding, it would need to be set up for even the control cases and a sham procedure (i.e. running of the instrument) should be done when salvaged blood will not be processed or transfused so that the blinded participants in the room will not know if it is actually being used or not.

Response: We thank the reviewer for this valuable suggestion. We agree that, in order to preserve blinding, the Cell Saver should be prepared and operated in both the intervention and control groups. Accordingly, we have revised the manuscript to clarify this point. The revised text now reads:

1. “In accordance with the instruction manual, we will prepare the Cell Saver Elite+ (Haemonetics, Boston, MA, USA) and install the cell salvage disposables in all cases, regardless of allocation to the intervention or control group, to perform the intraoperative cell salvage procedure.” (Page 10, Lines 158–161 in clean version)

2. “In the control group, the same collection, washing, and storage in the reinfusion bag will be performed as in the intervention group, in order to preserve blinding.” (Page 11–12, Lines 183–184 in clean version)

Point 2: Another point of clarification, on page 18, line 309, the first secondary outcome: This might be more of an exploratory outcome. For the study group, will you record how much of the RBC volume transfused is from cell salvage vs. banked RBCs? It would be interesting to see if there is a tendency to see more bleeding in cases in which a larger proportion of RBCs transfused comes from salvaged blood

Response: We thank the reviewer for this valuable comment. We selected intraoperative and early postoperative RBC transfusion volume as a secondary outcome because, if our hypothesis holds true that the use of salvaged RBCs does not increase bleeding compared with allogeneic transfusion, then the need for allogeneic RBCs should be reduced in the intervention group. Thus, this outcome is directly linked to verifying the clinical relevance of our hypothesis.

As suggested, we will also record, for the intervention group, the transfused RBC volume separately for cell-salvaged and allogeneic sources. Furthermore, we agree that it is of interest to examine whether cases receiving a larger proportion of salvaged RBCs show different bleeding tendencies. To address this point, we have already planned a subgroup analysis of the primary outcome according to the total volume of intraoperative salvaged and allogeneic RBC transfusions after weaning from CPB (≥ 800 mL vs. < 800 mL). We believe this will provide exploratory insights into the reviewer’s concern.

To clarify this in the manuscript, we have revised the Data collection and management section as follows:

“Other data, including baseline characteristics and all secondary outcomes, will be collected from the electronic database. Among these, the total RBC transfusion volume will be recorded for all patients, and in the intervention group, the transfused RBC volume will additionally be documented separately as cell-salvaged and allogeneic sources. All data will be entered into the respective CRF by unblinded researchers.” (Page 24–25, Lines 414–418 in clean version)

Responses to Reviewer #5

Comment: All suggestions have been considered. I believe the manuscript has improved in clarity and scientific value.

Response: We sincerely thank the reviewer for their positive evaluation and are grateful for their supportive comments.

---

## [Editor Report · Decision Letter 2]

29 Sep 2025

Intraoperative cell-salvaged versus allogeneic red blood cell transfusions in high-bleeding-risk cardiovascular surgery: Protocol for a single-center, randomized, parallel-group, noninferiority trial

PONE-D-25-26936R2

Dear Dr. Tsukinaga,

We’re pleased to inform you that your manuscript has been judged scientifically suitable for publication and will be formally accepted for publication once it meets all outstanding technical requirements.

Kind regards,

Mehmet Baysal

Academic Editor

PLOS ONE
---

## [Editor Report · Acceptance letter]

PONE-D-25-26936R2

PLOS ONE

Dear Dr. Tsukinaga,

I'm pleased to inform you that your manuscript has been deemed suitable for publication in PLOS ONE. Congratulations! Your manuscript is now being handed over to our production team.

Kind regards,

on behalf of

Dr. Mehmet Baysal

Academic Editor

PLOS ONE